# Secondary School Teachers and Outpatient Physicians: Differences in Attitudes towards Vaccination against COVID-19 in Slovakia

**DOI:** 10.3390/vaccines10111858

**Published:** 2022-11-02

**Authors:** Maria Tatarkova, Romana Ulbrichtova, Viera Svihrova, Jana Zibolenova, Martin Novak, Jan Svihra, Henrieta Hudeckova

**Affiliations:** 1Department of Public Health, Jessenius Faculty of Medicine in Martin, Comenius University in Bratislava, Mala Hora 11149/4B, 036 01 Martin, Slovakia; 2Clinic of Urology, Jessenius Faculty of Medicine in Martin, Comenius University in Bratislava, Kollarova 2, 036 59 Martin, Slovakia

**Keywords:** vaccination, influenza, COVID-19, outpatient physicians, teachers

## Abstract

The aim of this study was to evaluate the differences in attitudes towards vaccination against COVID-19 among secondary school teachers and outpatient physicians. A cross-sectional study was realised using anonymous questionnaires. The EPI Info 7 program and R software, version 4.0.2 were used for statistical analysis. The questionnaire was completed by 868 respondents (teaching staff N = 451; outpatient physician N = 417). The number of employees vaccinated against COVID-19 was 742 (85.5%). The number of those vaccinated against COVID-19 and influenza (last season) was 192 (21.9%). The statistically significant predictors were the level of fear of COVID-19 (OR 1.40; 95% CI 1.29–1.52), profession—outpatient physicians (OR 2.56; 95% CI 1.55–4.23), history of COVID-19 (OR 0.34; 95% CI 0.22–0.54), gender (OR 0.55; 95% CI 0.33–0.89) and influenza vaccination at any time in the past (OR 3.52; 95% CI 1.10–11.31). The strongest motivation for vaccination against COVID-19 among physicians was the prevention of the spread of COVID-19 during the performance of their profession (N = 336; 87%); among teachers, it was the protection of family members (N = 258; 73%). The most common reason for vaccine hesitancy was concern about vaccine safety (N = 80; 63.5%).

## 1. Introduction

Respiratory diseases are one of the main causes of death worldwide and are a major public health problem. An average of 389,000 deaths worldwide are associated with influenza each year, and more than 6.5 million deaths have been caused by COVID-19 [1]. We currently have vaccines available against both diseases. Nevertheless, vaccination against these diseases is low in some countries. Slovakia is undoubtedly one of the countries with low vaccination coverage, where only 12.8% of the at-risk population (population aged 65+) are vaccinated against influenza [2]. Slovakia is also among the countries with low vaccination rates against the COVID-19 disease. An example is a comparison with neighboring countries. The proportion of the population that is fully vaccinated against COVID-19 in Austria is more than 76%; in the Czech Republic and Hungary, it is approximately 64%; in Poland, it is almost 59%; and in Slovakia, almost 50% of the population is fully vaccinated [3].

The first case of COVID-19 was confirmed in Slovakia on 6 March 2020; five days later, the WHO declared a coronavirus pandemic [4]. Vaccination against COVID-19 began on 26 December 2020, while the national strategy of vaccination against COVID-19 in the conditions of the Slovak Republic divided the priority groups for the implementation of the vaccination into four groups (the first phase included healthcare workers, and the vaccine used was Pfizer/BioNTech; the second phase included people with chronic diseases and those 65+ years of age and the third phase included teachers (if they were not vaccinated according to the second phase), and the vaccine used was AstraZeneca). However, the study was carried out in the second half of 2021, when respondents could choose the type of vaccine. The first priority vaccination group included healthcare workers and teachers; they were vaccinated as a population group at a high risk of spreading the disease. Both professions—outpatient physicians and secondary school teaching staff—belong to the group of residents with a high risk of spreading COVID-19. Together, these professions can significantly influence the opinions and attitudes regarding the acceptance of a vaccination among the general population. Secondary school teachers can motivate and educate students on the importance of vaccination if their students ask them about this issue. Additionally, outpatient physicians can significantly influence their patients as a part of prevention. Consequently, we chose these professions as the target group in our investigation.

The aim of this study was to evaluate the differences in attitudes towards vaccination against COVID-19 among secondary school teachers and outpatient physicians.

## 2. Materials and Methods

### 2.1. Study Population

In the second half of 2021, a cross-sectional study was performed. The anonymous questionnaire was sent to the work emails of outpatient physicians (N = 1905) and secondary school teachers (N = 3401) working in the Žilina Region (Northern Region of Slovakia). The questionnaires were sent by the Žilina Self-Governing Region (Department of Education and Sport and Department of Health Care). Our sample size was estimated to be 345 for secondary school teachers and 319 for outpatient physicians. High-school teachers (who teach students aged 15–19) were included in the group of secondary school teachers.

This was calculated using a margin of error of 5 and based on the prevalence of COVID-19 vaccination. The response rate of the questionnaires among outpatient physicians was 21.9% (N = 417), and among secondary school teachers, it was 13.3% (N = 451). The respondents were instructed to complete the questionnaire only once (the avoidance of duplicate answers). Respondents who had been vaccinated with at last one dose of the COVID-19 vaccine were assigned to the vaccinated group. Respondents who were not vaccinated with any of the COVID-19 vaccines were included in the unvaccinated group. We also created a subgroup of vaccinated respondents which included respondents who were vaccinated with at last one dose of the COVID-19 vaccine and who were vaccinated against influenza last season. In this subgroup were respondents who were not vaccinated with any of the COVID-19 vaccines and have not been vaccinated against influenza.

### 2.2. Questionnaire

The modified questionnaire from Štěpánek et al. was used to carry out a cross-sectional study (the area of profession was changed from the original questionnaire) [5]. The questionnaire was created and distributed through Microsoft Forms (Office 365 online survey creator). The questionnaire consists of two parts, with the same questions for all respondents in the first part. The second part was divided according to vaccination against COVID-19 (different questions for vaccinated and unvaccinated respondents). The first part of the questionnaire consisted of questions focusing on the socio-demographic characteristics of the respondents. Most of the questions were closed. For both groups (vaccinated and unvaccinated), the same questions focused on gender, the number of years in the profession, job classification, the presence of chronic disease, influenza vaccination, the assessment of the general fear of COVID-19 (scale 0–10) and attitudes on mandatory vaccination against COVID-19 in different groups of the population (in my profession, selected groups and the general population) [6].

### 2.3. Data Analysis

The association between categorical variables was analysed using the chi-squared test and Fisher’s exact test. To compare the two averages, Student’s *t*-test was used. The EPI Info 7 program and R software, version 4.0.2 were used for multivariate logistic analysis (estimation of vaccination status against COVID-19 and influenza (last season)). Multivariate logistic regression was used to identify predictors of COVID-19 and influenza vaccination status. Odds ratios with 95% confidence intervals were used to identify statistically significant differences between the vaccinated and unvaccinated groups. A *p*-value < 0.05 was considered statistically significant.

## 3. Results

The questionnaire was completed by 868 respondents, of which 451 (52.0%) were teaching staff and 417 (48.0%) were outpatient physicians. The number of respondents vaccinated against COVID-19 was 742 (85.5%), of which 386 (52.0%) were outpatient physicians and 356 (48.0%) were teaching staff. The majority of the group vaccinated against COVID-19 (467; 62.9%) were vaccinated with Comirnaty (Pfizer a BioNTech), and 204 (27.5%) respondents were vaccinated with Vaxzevria (AstraZeneca). The Spikevax (Moderna) vaccine was administered to 66 (8.9%) respondents. The Janssen (Johnson & Johnson) vaccine was administered to four (0.5%) respondents, and the Sputnik V (Gamalej Institute) vaccine was administered to only one respondent (0.1%). The number of respondents vaccinated simultaneously (at once) against COVID-19 and seasonal influenza (last season) was 192 (21.9%); outpatient physicians made up 157 (82%) of the respondents and teachers made up 35 (18%) of the respondents. A statistically significant association was demonstrated between vaccination status against COVID-19 and profession, job duration, level of fear of COVID-19 (scale 0–10), history of COVID-19 and influenza vaccination (Table 1). A statistically significant association was demonstrated between vaccination status against COVID-19 and influenza (last season) and profession, job duration, level of fear of COVID-19, history of COVID-19, influenza vaccination at any time in the past and the presence of chronic disease (Table 2).

Multivariate logistic regression analysis was used to compare the vaccinated-against-COVID-19 group and the unvaccinated group. The predictors were gender, job duration in years, profession, history of COVID-19, the presence of a chronic disease, level of fear of COVID-19 and status of influenza vaccination. Gender, profession, history of COVID-19, level of fear of COVID-19 and being vaccinated against influenza at any time in the past were the factors positively associated with vaccination acceptance (Table 3).

Based on our results, the strongest motivations for vaccination were the prevention of the spread of COVID-19 during the performance of their profession (N = 571; 77.0%), the protection of family members (N = 543; 73.2%) and concerns about COVID-19 itself (N = 488; 65.8%). On the other side, the weakest motivation for vaccination was “Being exempted from restrictive anti-pandemic measures after vaccination” (N = 188; 25.3%). Between the professions, three questions were statistically significant, namely, “Concerns about COVID-19 itself”, “An effort to prevent the spread of COVID-19 during the performance of my profession” and “Being exempted from restrictive anti-pandemic measures after vaccination”; outpatient physicians reported these motivations more often than teaching staff (Table 4).

In the group that was unvaccinated against COVID-19 (N = 742), we looked for the reasons for vaccine hesitancy. The most common reason for hesitancy towards vaccination against COVID-19 was the concern about vaccine safety (N = 80; 63.5%); between professions, one question was statistically significant (the teaching staff were more concerned about the safety of vaccines compared to outpatient physicians). Only 28.6% (N = 36) of respondents who are not vaccinated against COVID-19 cite the presence of vaccination contraindications as the reason (Table 5).

In all areas, such as gender, vaccination against COVID-19, number of years in the profession, being vaccinated against influenza at any time in the past and chronic diseases (with or without), no statistical significance was noted. However, statistical significance was noted for influenza vaccination (last season), and teaching staff were more concerned about COVID-19 compared to outpatient physicians (Table 6).

Respondents were asked for their opinions on mandatory vaccination against COVID-19 in selected groups of the population. When comparing by profession, as well as when comparing by vaccination status against COVID-19, all areas were statistically significant. Outpatient physicians more often expressed a positive opinion about mandatory vaccination compared to teaching staff. The COVID-19-vaccinated group expressed significantly higher agreement with mandatory vaccination compared to the COVID-19-unvaccinated group (Table 7).

Multivariate logistic regression analysis was used to compare vaccinated (simultaneously against COVID-19 and seasonal influenza (last season)) and unvaccinated or partially vaccinated respondents. The predictors were gender, job duration in years (>40; 20–39 years), job classification (outpatient physicians vs. teaching staff), the presence of a chronic disease, level of fear of COVID-19 and history of COVID-19. Profession, fear of COVID-19 and job duration in years (more than 40) were factors that were positively associated with vaccination acceptance. Predictors such as gender, job duration (20–39 years) and present chronic disease were negatively associated (Figure 1).

## 4. Discussion

Outpatient physicians and teachers represent a risk group that may be more exposed to respiratory diseases such as COVID-19 or influenza. Vaccination against COVID-19 is not mandatory in Slovakia. The acceptance of vaccination against COVID-19 among teachers is variable between countries. Countries such as Portugal (100% vaccination rate), Chile (97%), Sweden (97%) and Saudi Arabia (96%) have the highest vaccination rates against COVID-19 among teachers. It is interesting that, despite the introduction of mandatory vaccination in Uzbekistan, only 60% of teachers in this country were fully vaccinated by the end of 2021 (October) [7]. In our sample (including outpatient physicians; N = 868), 48% of teachers were vaccinated against COVID-19 with at least one dose, and 75% of respondents who were not vaccinated against COVID-19 were teachers. However, if we were to exclude outpatient physicians from the entire sample of respondents and evaluate only teachers (N = 417), vaccinated teachers made up 78.9% (N = 356) and unvaccinated teacher made up 21.1% (N = 95). Only a few studies have assessed the attitudes towards and reasons for hesitating to receive a COVID-19 vaccination among teachers. In a study by King et al. (2021), hesitancy to accept the COVID-19 vaccination was different between high school teachers (3.6%) and preschool/kindergarten teachers (14.8%) [8]. A study from China assessed the hesitancy to get vaccinated against COVID-19 among teachers and students. The hesitancy to get vaccinated against COVID-19 was 37.1% among teachers and 23.8% among students [9]. In our study, 21.1% of teachers hesitated to get vaccinated against COVID-19. The most common reasons for hesitancy among teachers were the concern about the safety and side effects of vaccines against COVID-19 (68%; N = 65) and the distrust of the effectiveness of vaccines against COVID-19 (40%; N = 38). Concern about the safety and side effects of vaccines against COVID-19 is common not only among teachers or healthcare workers but also among the general population [5,6,10,11].

In a study by Gesser-Edelsburg et al., 81% of physicians were vaccinated or planned to be vaccinated against COVID-19, while 8% of physicians hesitated to be vaccinated, and 5% said they refused vaccination [12]. In the studies of Parsons Leigh et al., 85.6% of physicians were vaccinated, while 6.5% of physicians reported being hesitant to get vaccinated against COVID-19. An interesting finding from this study is that physicians in high-income countries reported a greater hesitancy to receive the COVID-19 vaccine compared to physicians in low- and middle-income countries [13]. In a study by Callaghan et al., it was found that 5.2% of primary care physicians were unvaccinated, with 67.4% of primary care physicians strongly agreeing that vaccines are safe, 75% of primary care physicians agreeing strongly that vaccines are effective and 76.3% of primary care physicians agreeing that vaccines against COVID-19 are important [14]. In our study, 7.4% (N = 31) of outpatient physicians were hesitant to get vaccinated against COVID-19; this finding is also comparable with the mentioned studies. The main reason was concern about the safety and side effects of vaccines against COVID-19 (48.4%). Concerns about vaccine safety among physicians have been identified as a barrier to vaccine uptake in several studies, including our study [12,13].

Influenza vaccination has been available for many years, but in some countries, including Slovakia, a relatively small percentage of the at-risk population is vaccinated against influenza. For example, 22.3% of the at-risk population is vaccinated in Hungary, and the percentage is 23.9% in the Czech Republic. Germany has 47.3% of the population aged 65+ vaccinated, and Slovakia only has 12.8% [2]. Few studies have looked at teachers’ attitudes towards vaccination against this respiratory disease [15]. From the available sources, it is evident that teachers have insufficient knowledge about influenza and often have misconceptions that influenza is like a cold and that only older age groups and the immunocompromised need a vaccination against it. According to a study by Vaughn et al. (2018), which dealt with teachers’ knowledge and attitudes about influenza, only 35.9% of respondents reported annual influenza vaccination in the past 5 years, and up to 17% of respondents reported not having received any influenza vaccine in the past 5 years [16]. Similar results were also recorded in a Greek study, where 34.8% of teachers reported influenza vaccination in the 2019/2020 season [17]. Disturbing results were recorded in the Polish study, where only 4.7% of teachers were vaccinated against influenza in the 2018/2019 season, while 24.5% of teachers reported that they had been vaccinated against influenza sometime in previous seasons [18]. Our results are similar to the results of a study conducted in the Czech Republic, with the vaccination rate of respondents being 6% [19]. The results of our study show that only 7.8% of teachers were vaccinated against influenza in the last season, while 15.7% reported having an influenza vaccination sometime in the past, and up to 76.5% of teachers reported that they had never been vaccinated against influenza. It should be noted that 22.6% (N = 102) of the entire group of teachers reported a chronic disease, up to 60.8% (N = 62) of teachers with a chronic disease reported that they had never been vaccinated against influenza and 88.2% (N = 90) of teachers with a chronic disease had not been vaccinated against influenza in the last season.

The vaccination rates of physicians against seasonal influenza are relatively variable in different studies. This may be due to the different selection of the sample (hospital, family physicians or in the medical profession in general), similar to what some studies have reported with vaccinations against COVID-19 [20]. In a study from France, 55% of physicians were vaccinated against influenza [21]. In Poland, more than 32% of physicians were vaccinated in the 2016/2017 season [18]. A Slovenian study recorded 27% influenza vaccination among family physicians, and this result also correlated with a study from Turkey, where 27.3% of family physicians are annually vaccinated against influenza [22,23]. In our study, 38.6% of outpatient physicians were vaccinated against influenza in the last season, but up to 43.6% of outpatient physicians had never been vaccinated against influenza. It should be noted that the risk groups for severe influenza are people with the presence of a chronic disease. In our study, up to 37.2% (N = 58) of outpatient physicians with the presence of at least one chronic disease had never been vaccinated against influenza, and 53.8% (N = 84) of outpatient physicians with a chronic disease were not vaccinated against influenza in the last season. When comparing influenza vaccination between teachers and outpatient physicians, it is obvious that teachers accept vaccination at a significantly lower rate compared to physicians, not only in Slovakia but also in other countries where similar studies have been performed. According to available studies, the most significant positive predictor for the acceptance of the COVID-19 vaccine is influenza vaccination. People who are vaccinated against seasonal influenza have a positive attitude towards vaccination against COVID-19 [5,6,24]. This relationship was also recorded in our study, where 194 respondents were vaccinated in the last season, while 192 respondents were also vaccinated against COVID-19. As in other studies, our results confirm that physicians are more often vaccinated against seasonal influenza and COVID-19 compared to teaching staff.

### Limitations and Strengths

Our study has several limitations. It is a cross-sectional study with specific population groups, and the study analyses the situation in only one of the eight regions of Slovakia. Another limitation is due to the use of a non-standardised questionnaire and the relatively low return rate of the questionnaire, especially among secondary school teachers. On the other hand, the sample size was estimated to be 345 for secondary school teachers and 319 for outpatient physicians, which is slightly lower than the size of our sample in both monitored groups of respondents. A major strength is the fact that there are relatively few studies that assess the differences in attitudes towards vaccination against COVID-19 between outpatient physicians and secondary school teaching staff. For this reason, we consider our results to be an input to this field.

## 5. Conclusions

Outpatient physicians and secondary school teaching staff belong to the population group with a high risk of spreading the COVID-19 disease. Additionally, it is a population group that is exposed to a greater risk for influenza. The most effective prevention against these diseases is vaccination. The most significant positive predictor for the acceptance of the COVID-19 vaccine is influenza vaccination (last season). The strongest motivations for vaccination against COVID-19 were the prevention of the spread of COVID-19 during the performance of the profession (dominant among outpatient physicians) and the protection of family members (dominant among teachers). The most common reason for vaccine hesitancy against COVID-19 was the concern about vaccine safety (both professions). Teachers were more likely to be hesitant to get vaccinated against COVID-19 and influenza compared to outpatient physicians.

Outpatient physicians and secondary school teaching staff play a significant role in influencing the opinions of the population. From a public health standpoint, it is necessary to know the reasons for refusing vaccination. Defining barriers will enable targeted education, which can have a significantly positive influence on the acceptance of vaccination.

## Figures and Tables

**Figure 1 vaccines-10-01858-f001:**
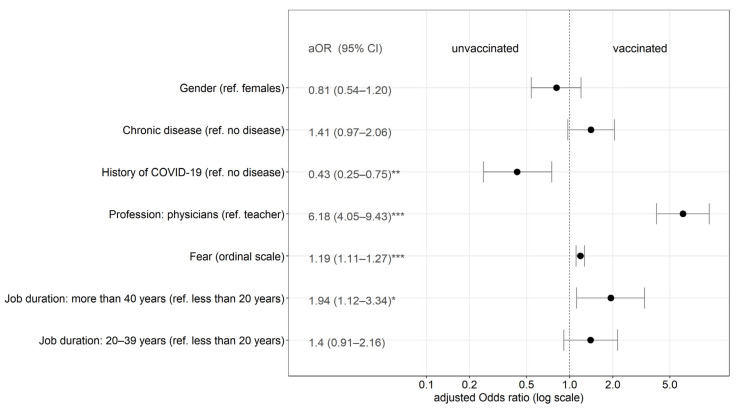
Predictors of simultaneous vaccination against COVID-19 and seasonal influenza (last season) (multivariate logistic regression analysis). Note: * *p* < 0.05; ** *p* < 0.01; *** *p* < 0.001.

**Table 1 vaccines-10-01858-t001:** Socio-demographic characteristics of respondents vaccinated against COVID-19.

Variables	Total (N = 868)	Vaccinated (N = 742)	Unvaccinated (N = 126)	*p* Value
Male N (%)	261 (30%)	230 (31%)	31 (25%)	0.15
Female N (%)	607 (70%)	512 (69%)	95 (75%)	
Teaching staff N (%)	451 (52%)	356 (48%)	95 (75%)	<0.001 **
Outpatient doctors N (%)	417 (48%)	386 (52%)	31(25%)	
Job duration (average years ± SD)	23.81 ± 12.55	24.39 ± 12.62	20.36 ± 11.67	<0.001 *
Level of fear of COVID-19 (average ± SD)	5.98 ± 2.82	6.33 ± 2.68	3.90 ± 2.73	<0.001 *
History of COVID-19 N (%)	168 (19%)	120 (16%)	48 (38%)	<0.001 **
Vaccinated against influenza at any time in the past N (%)	341 (39%)	318 (43%)	23 (18%)	<0.001 **
Vaccinated against influenza last season N (%)	194 (22%)	190 (26%)	4 (3%)	<0.001 **
With a chronic disease N (%)	258 (30%)	228 (31%)	30 (24%)	0.12

* *p* < 0.05 (Student’s *t*-test); ** *p* <0.05 (chi-square test).

**Table 2 vaccines-10-01858-t002:** Socio-demographic characteristics of respondents simultaneously vaccinated against COVID-19 and influenza (last season).

Variables	Total (N = 868)	Vaccinated (N = 192)	Unvaccinated (N = 676)	*p* Value
Male N (%)	261 (30%)	58 (30%)	203 (30%)	0.96
Female N (%)	607 (70%)	134 (70%)	473 (70%)	
Teaching staff N (%)	451 (52%)	35 (18%)	416 (62%)	<0.001 **
Outpatient physicians N (%)	417 (48%)	157 (82%)	260 (39%)	
Job duration (average years ± SD)	23.81 ± 12.55	28.88 ± 12.08	22.37 ± 12.33	<0.001 *
Level of fear of COVID-19 (average ± SD)	5.98 ± 2.82	6.95 ± 2.51	5.70 ± 2.85	<0.001 *
History of COVID-19 N (%)	168 (19%)	18 (20%)	150 (22%)	<0.001 **
Vaccinated against influenza at any time in the past N (%)	341 (39%)	192 (100%)	149 (22%)	<0.001 **
With a chronic disease N (%)	258 (30%)	81 (42%)	177 (26%)	<0.001 **

Note: SD, standard deviation; * *p* < 0.05 (Student’s *t*-test); ** *p* < 0.05 (chi-square Test; Fisher exact test).

**Table 3 vaccines-10-01858-t003:** Predictors of COVID-19 vaccination (multivariate logistic regression analysis).

Variable	Odds Ratio	95% CI	*p* Value
Gender (females)	0.55	0.33–0.89	<0.05 *
Job duration (years)	1.01	0.99–1.03	0.38
Profession (physicians)	2.56	1.55–4.23	<0.001 **
History of COVID-19	0.34	0.22–0.54	<0.001 **
With a chronic disease	0.73	0.44–1.22	0.23
Level of fear of COVID-19	1.40	1.29–1.52	<0.001 **
Vaccinated against influenza (last season)	1.59	0.88–2.87	0.12
Vaccinated against influenza (past)	3.52	1.10–11.31	<0.05 *

Note: CI: confidence interval; * *p* < 0.05; ** *p* < 0.001.

**Table 4 vaccines-10-01858-t004:** Motives to get vaccinated against COVID-19 (N = 742).

Motives	Teaching Staff (N = 356)	Outpatient Physicians (N = 386)	*p* Value
Concerns about COVID-19 itself	213 (60%)	275 (71%)	<0.01 *
An effort to prevent the spread of COVID-19 during the performance of my profession	235 (66%)	336 (87%)	<0.001 *
An effort to protect family members	258 (73%)	285 (74%)	0.68
Being exempted from restrictive anti-pandemic measures after vaccination	77 (22%)	111 (29%)	<0.05 *

* *p* < 0.05 (chi-square test); multiple-choice options.

**Table 5 vaccines-10-01858-t005:** Reasons for hesitating to get vaccinated against COVID-19 (N = 126).

Reason for Hesitation	Teaching Staff (N = 95)	Outpatient Physicians (N = 31)	*p* Value
I am not afraid of COVID-19, its course and consequences	9 (10%)	4 (13%)	0.59
I do not find getting infected with COVID-19 likely	4 (4%)	0	0.57
I do not trust the efficacy of vaccines against COVID-19	38 (40%)	7 (23%)	0.08
I have concerns about the safety and side effects of vaccines against COVID-19	65 (68%)	15 (48%)	<0.05 *
I went through COVID-19 and assume lasting immunity against the disease	16 (17%)	10 (32%)	0.07
I have contraindications and expect a complicated vaccination course in my case	27 (28%)	9 (29%)	0.95

* *p* < 0.05 (chi-square test; Fisher exact test); multiple-choice options.

**Table 6 vaccines-10-01858-t006:** Level of fear of COVID-19 among respondents.

	Teaching Staff (Mean ± SD)	Outpatient Physicians (Mean ± SD)	*p* Value
Females	6.17 ± 2.80	6.29 ± 2.74	0.59
Males	5.32 ± 2.81	5.46 ± 2.92	0.70
Vaccinated	6.45 ± 2.60	6.22 ± 2.76	0.24
Unvaccinated	3.94 ± 2.78	3.77 ± 2.63	0.77
Job duration	5.92 ± 2.83	6.04 ± 2.82	0.55
Vaccinated against influenza at any time in the past N (%)	6.17 ± 2.74	6.37 ± 2.79	0.53
Vaccinated against influenza last season N (%)	7.74 ± 2.09	6.68 ± 2.63	<0.05 *
With a chronic disease	6.75 ± 2.57	6.53 ± 2.76	0.52
Without a chronic disease	5.68 ± 2.86	5.75 ± 2.82	0.78

Note: SD, standard deviation; * *p* < 0.05 (Student’s *t*-test).

**Table 7 vaccines-10-01858-t007:** Opinion on the introduction of mandatory vaccination against COVID-19.

Occupation				COVID-19 Vaccination		
Mandatory Vaccination	Teaching Staff (N = 451)	Outpatient Physicians (N = 417)	*p* Value	Vaccinated (N = 742)	Unvaccinated (N = 126)	*p* Value
In my profession	163 (36%)	274 (66%)	<0.001 *	436 (59%)	1 (0.8%)	<0.001 *
Selected population	172 (38%)	275 (66%)	<0.001 *	442 (60%)	5 (4%)	<0.001 *
Whole population	110 (24%)	176 (42%)	<0.001 *	285 (28%)	1 (0.8%)	<0.001 *

* *p* < 0.05 (chi-square Test).

## Data Availability

All data are fully available without any restriction upon reasonable request.

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
