# Peer review of "Secondary School Teachers and Outpatient Physicians: Differences in Attitudes towards Vaccination against COVID-19 in Slovakia"

_vaccines, 2022, doi:10.3390/vaccines10111858_

Round 1

Reviewer 1 Report

The manuscript led by Tatarkova et al entitled “secondary teachers and outpatient physicians: differences in attitudes towards vaccination against COVID-19” is very interesting and the author explored novel aspects of attitude and vaccination. The author documented this manuscript very well and discuss the results very interestingly. So this manuscript is suitable for publication in Vaccines journals. However, the author needs to address the following minor comments before to accepted for publication

Comments to author

The physician's attitude directly impact vaccination but not clear how especially secondary teachers impact it. The impact of secondary teachers is more than parents’ impact on vaccination

Line 40: WHO declared a coronavirus pandemic. The author should cite https://www.sciencedirect.com/science/article/pii/S0169409X21000028

 The secondary teacher's definition is not clear in the manuscript. Each country has different classes in secondary. Which class teachers the author considered as secondary teachers in this manuscript need to be clear

Suggested to the author to make number to tables as Table 1, 2, 3, 4, and so on instead of 1a and 1b, and 2 and so on

In the table, the author has a column vaccinated. Which are the vaccines author considered?

Line 121: Note: CI confidence interval should be Note: CI: confidence interval

Citation and references are not in sequence. Line 167 has 20-39 and the next section Discussion starts the references number from 6.

Strongly discouraging to use 20-39. Almost 20 citations together. The author should limit the number to 1 or 2. and there is no demand at all to cite 20 references. The author should be very careful

The quality of figure 1 is poor. The author should represent it a more interesting way for readers

Author Response

Response to Reviewer 1 Comments

Dear reviewer, 

thank you for your comments, which were very inspiring for us. We have edited the text of the manuscript according to your recommendations.

Point 1: “The physician's attitude directly impact vaccination but not clear how especially secondary teachers impact it. The impact of secondary teachers is more than parents’ impact on vaccination.”

Response 1:

Thank you for your comment, we also agree with you. Secondary school teachers can motivate and educate students on the importance of vaccination, if students ask them about this issue.

We also mentioned it in our manuscript: “Secondary school teachers can motivate and educate students on the importance of vaccination, if their students ask them about this issue”.

Point 2: Line 40: WHO declared a coronavirus pandemic. The author should cite https://www.sciencedirect.com/science/article/pii/S0169409X21000028

Response 2: We have added a cite.

Point 3: The secondary teacher's definition is not clear in the manuscript. Each country has different classes in secondary. Which class teachers the author considered as secondary teachers in this manuscript need to be clear.

Response 3: We added: High-school teachers (who teach students aged 15-19) were included in the group of secondary teachers.

Point 4: Suggested to the author to make number to tables as Table 1, 2, 3, 4, and so on instead of 1a and 1b, and 2 and so on.

Response 4: Thank you for your comment. Tables 1a and 1b present the socio-demographic characteristics of the respondents. For this reason, we chose these types (preventing confusion when assigning respondents).

Point 5: In the table, the author has a column vaccinated. Which are the vaccines author considered?

Response 5: We added: “The number of respondents vaccinated against COVID-19 was 742 (85.5%), of which 386 (52.0%) outpatient physicians and 356 (48.0%) teaching staff. The majority of the vaccinated group against COVID-19 (467; 62.9%) were vaccinated with Comirnaty (Pfizer a BioNTech), 204 (27.5%) respondents were vaccinated with Vaxzevria (AstraZeneca). The Spikevax (Moderna) was used in 66 (8.9%). The Janssen (Johnson&Johnson) was used in 4 (0.5%) and the Sputnik V (Gamalej Institute) was applied to only one respondent (0.1%).”

Point 6: Line 121: Note: CI confidence interval should be Note: CI: confidence interval

Response 6: We edited as recommended.

Point 7: Citation and references are not in sequence. Line 167 has 20-39 and the next section Discussion starts the references number from 6.

Response 7: Thank you for your comment. Numbers are not references- represent job duration in years. We added "years" to the numbers in brackets.

Point 8:

Strongly discouraging to use 20-39. Almost 20 citations together. The author should limit the number to 1 or 2. and there is no demand at all to cite 20 references. The author should be very careful.

Response 8: Thank you for your comment. Numbers are not references- represent job duration in years. We added "years" to the numbers in brackets.

Point 9:

The quality of figure 1 is poor. The author should represent it a more interesting way for readers

Response 9: The image was created in R software, version 4.0.2 were used for statistical analysis.  The reduced quality of the image may be caused by inserting it into a Microsoft Word template (accepted file formats).

Yours sincerely

Dr. Romana Ulbrichtova

Reviewer 2 Report

The title should be changed to include the words "in Slovakia".   The Discussion section wanders, without building any clear messages.  Are you looking for things you can change to achieve more vaccination or just describing why people don't get vaccinated in Slovakia?  Your work shows that people who respond to influenza vaccination campaigns are also likely to trust to new vaccines.  Perhaps a key point is that it may take different measures to convince teachers of young children compared to older students.   Also, what was the point of discussing variations in vaccination rates between countries, by grade taught and income?  We would expect variation due to sampling alone.  Not clear what trends are observed by country.  Differences between high income countries may signal that people feel more resistant to disease just because of their wealth.

Author Response

Response to Reviewer 2 Comments

Dear reviewer, 

thank you for your comments, which were very inspiring for us. We have edited the text of the manuscript according to your recommendations.

Point 1: The title should be changed to include the words "in Slovakia".    

Response 1: Thank you for your comment. "in Slovakia" was added to the title

Point 2: “The Discussion section wanders, without building any clear messages.  Are you looking for things you can change to achieve more vaccination or just describing why people don't get vaccinated in Slovakia??”

Response 2: The aim of this study was to evaluate differences towards attitudes towards vaccination against COVID-19 among secondary teachers and outpatient physicians. We chose this aim because there were enough vaccines during the study period, but people hesitated or refused vaccination.

Point 3: Your work shows that people who respond to influenza vaccination campaigns are also likely to trust to new vaccines.  Perhaps a key point is that it may take different measures to convince teachers of young children compared to older students.

Response 3: Thank you for your comment. In our study, we focused only on the evaluation of the attitudes towards vaccination among outpatient physicians and secondary teachers.

Point 4: Also, what was the point of discussing variations in vaccination rates between countries, by grade taught and income?  We would expect variation due to sampling alone.  Not clear what trends are observed by country.  Differences between high income countries may signal that people feel more resistant to disease just because of their wealth.

Response 4:

We discussed the differences in vaccination rates because of the effort to get closer to the current situation in different countries. The aim was not to analyse the differences in vaccination between countries due to their economic situation.

Yours sincerely

Dr. Romana Ulbrichtova

Reviewer 3 Report

Dear authors it is  an interesting and well written article

Some comments

Introduction

Was the same vaccine offered for booth groups? (E.g., Pfizer sometimes is more accepted than Astra Zeneca) if not could this influence in the vaccination decision?

Materials and Methods

Who has provided the email of the teachers and physicians?

Was the questionnaire send to all of them?

Results

Table 1

How do you measure COVID-19 19 fear? Scale 0 to 10?

Was mandatory vaccination for booth groups at some point?

Author Response

Response to Reviewer 3 Comments

Dear reviewer, 

thank you for your comments, which were very inspiring for us. We have edited the text of the manuscript according to your recommendations.

Point 1: “Introduction: Was the same vaccine offered for booth groups? (E.g., Pfizer sometimes is more accepted than Astra Zeneca) if not could this influence in the vaccination decision?”

Response 1: We added: Vaccination against COVID-19 began on December 26, 2020, while the national strategy of vaccination against the disease COVID-19 in the conditions of the Slovak Republic divided the priority groups for the implementation of the vaccination into 4 groups (the first phase included health care workers, vaccine used: Pfizer/BioNTech; the second phase included people with chronic diseases and 65+, and the third phase included teachers (if they were not vaccinated according to the second phase) - vaccine used: AstraZeneca). However, the study was carried out in the second half of 2021, when respondents could choose the type of vaccine.

Point 2: “Materials and Methods: Who has provided the email of the teachers and physicians?

Was the questionnaire send to all of them?”

Response 2: We added: The questionnaires were sent by the Žilina Self-Governing Region (Department of Education and Sport and Department of Health Care).

Point 3: “Results: Table 1- How do you measure COVID-19 19 fear? Scale 0 to 10?

Was mandatory vaccination for booth groups at some point?”

Response 3: We added: “level of fear of COVID-19 (scale 0-10)”. We added to discussion: “Vaccination against COVID-19 is not mandatory in Slovakia”.

Yours sincerely

Dr. Romana Ulbrichtova
